# Effectiveness of Mesenchymal Stem Cell Therapy for COVID-19-Induced ARDS Patients: A Case Report

**DOI:** 10.3390/medicina58121698

**Published:** 2022-11-22

**Authors:** Kwangmin Kim, Keum Seok Bae, Hyun Soo Kim, Won-Yeon Lee

**Affiliations:** 1Department of Surgery, Yonsei University Wonju College of Medicine, Wonju 26426, Republic of Korea; 2Pharmicell Co., Ltd., Sungnam 13229, Republic of Korea; 3Kim’s Stem Cell Clinic, Seoul 06017, Republic of Korea; 4Department of Pulmonology, Yonsei University Wonju College of Medicine, Wonju 26426, Republic of Korea

**Keywords:** coronavirus disease 2019, acute respiratory distress syndrome, human bone marrow-derived mesenchymal stem cell

## Abstract

*Purpose:* This study assessed the safety, feasibility, and tolerability of mesenchymal stem cells for patients diagnosed with COVID (Coronavirus disease 2019-induced ARDS (acute respiratory distress syndrome)). *Materials and Methods:* Critically ill adult COVID-19 patients who were admitted to Wonju Severance Christian Hospital were enrolled in this study. One patient received human bone marrow-derived mesenchymal stem cell (hBMSC) transplantation and received a total dose of 9 × 10^7^ allogeneic hBMSCs via intravenous infusion. The main outcome of this study was to assess the safety, adverse events, and efficacy following transplantation of hBMSCs in COVID-19- induced ARDS patients. Efficacy was assessed radiologically based on pneumonia improvement, changes in PaO_2_/FiO_2_, and O_2_ saturation. *Results:* A 73-year-old man visited Wonju Severance Christian Hospital presenting with fever and fatigue. A throat swab was performed for real-time polymerase chain reaction to confirm COVID-19, and the result was positive. The patient developed ARDS on Day 5. MSC transplantation was performed on that day and administered on Day 29. Early adverse events, including allergic reactions, were not observed following MSC transplantation. Subsequently, clinical symptoms, signs, and laboratory findings, including PaO_2_/FiO_2_ and O_2_ saturation, improved. *Conclusion:* The results of this case report suggest that intravenous injection of MSC derived from the bone marrow is safe and acceptable and can lead to favorable outcomes for critically ill COVID-19 patients.

## 1. Introduction

Acute respiratory distress syndrome (ARDS) is a life-threatening illness and is the primary cause of death and disability among critically ill patients; the mortality rate ranges from 26–50% [1,2,3]. ARDS is an inflammatory condition characterized by the infiltration of mixed inflammatory cells, disruption of the alveolar–capillary barrier, severe edema, hypoxemia, and increased lung density [4].

The causes of ARDS include bacterial or viral pneumonia and severe traumatic lung injury. ARDS has been managed conservatively, including mechanical ventilation and treatment of the underlying disease. Despite these treatments, ARDS is usually associated with a high mortality rate. Previous experimental investigations indicate that intra-tracheal or intravenous administration of bone marrow-derived stem cells can reduce lung inflammation, accelerate lung repair, restore alveolar fluid clearance, and improve arterial oxygenation [5,6,7].

Coronavirus disease 2019 (COVID-19) is an infectious disease caused by severe acute respiratory syndrome coronavirus 2 (SARS-CoV-2), initially reported in Wuhan, Hubei, China, in December 2019 [8]. COVID-19 has a wide range of respiratory and non-respiratory clinical manifestations, including mild or severe flu-like symptoms, pneumonia, and ARDS, which when left untreated may result in sepsis and multi-organ distress syndrome. Respiratory failure related to ARDS is the leading cause of intensive care unit (ICU) admissions [9,10]. Symptomatic therapy is adequate for patients with mild COVID-19 symptoms, but there is still no definitive treatment for patients with severe symptoms. Different therapeutic approaches, such as hydroxychloroquine, azithromycin, lopinavir, and veklury, have been evaluated with no conclusive results [11].

Recently, some studies suggest that cell-based therapy provides therapeutic effects for critically ill patients with pneumonia due to COVID-19 [12,13]. The role of these stem cells may be supporting hematopoietic function, cell regeneration using multipotent cell differentiation ability, and immune modulation [14]. Therefore, it may be possible to suppress the overexpressed inflammatory response in ARDS patients.

We hypothesized that stem cell therapy can be effective for patients with ARDS caused by COVID-19. Therefore, the purpose of this study was to evaluate the safety, feasibility, and tolerance of mesenchymal stem cells in COVID-19-induced ARDS patients.

## 2. Materials and Methods

### 2.1. Patient Eligibility

Adult COVID-19 patients in critical condition admitted to Wonju Severance Christian Hospital were enrolled in this study. COVID-19 infection was confirmed using reverse transcription polymerase chain reaction (RT-PCR). The inclusion criteria were as follows: (1) 50–90 years old; (2) patients with pneumonia confirmed by chest radiography or computed tomography (CT); (3) patients with a respiratory rate of 30 breaths or more, severe dyspnea, or <90% oxygen saturation; (4) patients diagnosed with ARDS per WHO guidelines for definition and classification of ARDS [15]; and (5) patients who voluntarily consented to enroll in this study. The exclusion criteria included: (1) patients with a history of malignancy; (2) patients with a marked decrease in liver or renal function; (3) patients diagnosed with human immunodeficiency virus (HIV); (4) patients who had been diagnosed with venous thromboembolism or pulmonary arterial hypertension within 3 months prior to enrollment; (5) patients who had received immunosuppressive drugs for more than 1 month prior to registration; (6) patients with surgery, organ biopsy, or significant trauma within 3 months prior to enrollment; (7) patients who participated in other clinical trials within 1 month prior to enrollment; (8) patients who received cell therapy prior to enrollment; and (9) patients who were evaluated by the physician as inappropriate to participate in this trial.

### 2.2. Mesenchymal Stem Cell (MSC) Preparation and Infusion

The Institutional Review Board of the Public Institutional Bioethics Committee (Ministry of Health and Welfare, Seoul, Republic of Korea [P01-201901-31-002]) approved the study protocol which was performed according to the Declaration of Helsinki. Samples of bone marrow were taken from healthy volunteers. All production and testing processes for human bone marrow-derived mesenchymal stem cells (hBMSC) were conducted per appropriate manufacturing practices (Pharmicell Co. Ltd., Seongnam, Gyeonggi, Republic of Korea). Bone marrow mononuclear cells were extracted using density gradient centrifugation (Ficoll-Paque, 1.077 g/L, Sigma, St. Louis, MO, USA) and phosphate-buffered saline washing (PBS, Thermo Scientific, Waltham, MA, USA). The cells were resuspended in Dulbecco’s modified Eagle’s medium-low glucose (Gibco, Hopkinton, MA, USA) with 10% PLUSTM human platelet lysate (hPL) (Compass Biomedical, Hopkinton, MA, USA) and 20 g/mL gentamicin (Gibco). The cells were plated at a density of 1.0–1.5 × 10^5^ cells/cm^2^ in T-75 flasks. The cultures were maintained at 37 °C in a humidified environment containing 5% carbon dioxide. Non-adherent cells were eliminated by changing the media after 5 days, and adherent cells were cultivated for an additional 3 days (passage 1). Upon achieving 70–80 percent confluence, adherent cells were separated using TripLE Express (Gibco) and replated at a density of 4–5 × 10^3^ cells/cm^2^ in T-175 flasks. To prepare a Master Cell Bank, cultured cells (passage 2) were harvested, frozen using a cryostabilizer (CELLBANKER^®^2, ZENOAQ, Koriyama, Japan), and kept in a vapor phase liquid nitrogen tank (MCB, Cryotherm, Germany). Cells were thawed from the MCB, cultivated for 3–5 days (passage 3–6), harvested from passage 6, and then frozen to prepare the Working Cell Bank (WCB). WCB cells were thawed and cultured until passage 8. Eighth passage cells were cultured in a hypoxia-conditioned incubator (37 °C, 1% O_2_, 5% CO_2_, and 94% N_2_) for 72 h. Following hypoxic preconditioning, hBMSCs were collected and washed twice with PBS and once with Plasma Solution A inj. (Multiple Electrolytes Injection, Type 1, USP; HKinno.N crop, Thermo Scientific, Waltham, MA, USA). hBMSCs were resuspended in Plasma Solution A inj. to a final concentration of 9 × 10^7^ cells. The release of hBMSCs for clinical use was contingent upon the absence of microbial contamination (bacteria, fungi, virus, and mycoplasma), 80% viability in a trypan blue exclusion assay, 0.5 EU/mL endotoxin level, and immune phenotyping via flow cytometric analysis, demonstrating the expression of CD73 and CD105 surface markers and the absence of CD14, CD34, CD45, and HLA-DR.

One patient received hBMSC transplantation. The patient was administered 9 × 10^7^ allogeneic hBMSCs via an intravenous route. The infusion time was approximately 50 min. The patient received standard medications according to the patient’s condition.

### 2.3. Follow-Up for Patient

Vital signs measurement, chest radiography, and laboratory tests, including arterial blood gas analysis (ABGA), chemistry, complete blood counts (CBC), and urine analysis, were performed on days 0, 1, 3, 5, 7, 14, 21, and 28 after administration. Chest CT was performed on days 3, 14, and 28 after administration, if the patients’ condition was tolerable.

### 2.4. Outcome Measurement

The primary outcomes of this study were the safety, side effects, and effectiveness of hBMSC transplantation in COVID-19-induced ARDS patients. Common Terminology Criteria for Adverse Events (CTCAE) (version 5.0) are used to describe the severity of side effects [16]. Early side effects were defined as follows: allergic reactions including maculopapular rashes or urticaria without fever or hypotension; severely manifested with anaphylactic reactions that manifest as worsening of dyspnea, wheezing, anxiety, hypotension without fever, bronchospasms, cell embolism in the lungs or, less frequently, deteriorated organ function caused by large aggregations of cells in the heart during the IV infusion. This study was terminated when severe anaphylactic reactions or embolism occurred. Efficacy was assessed radiologically based on pneumonia improvement, changes in PaO_2_/FiO_2_, and O_2_ saturation.

## 3. Case Presentation

On 3 March 2020, A 73-year-old man with a history of diabetes mellitus, hypertension, cerebral infarction, and hyperlipidemia visited our emergency department with fever and fatigue. His initial blood pressure and pulse rate were within normal ranges and his temperature was 37.7 °C. Physical examination revealed no specific signs. Lung auscultation revealed normal breathing sounds, and chest radiography showed ground-glass opacities in both lower lung fields. Laboratory tests showed normal leukocyte (5.14 × 10^9^/L), lymphocyte (1.34 × 10^9^/L), elevated procalcitonin level (0.07 ng/mL), high C-reactive protein (CRP) (7.16 mg/dL), and creatinine (1.41 mg/dL) levels. A throat swab was performed for RT-PCR to confirm COVID-19, and the result was positive. Treatment with chloroquine and lopinavir/ritonavir was initiated.

On Day 4 of hospitalization, the patient developed a fever of 38.6 °C shortness of breath requiring oxygen supplementation via nasal high flow system (flow rate: 40 L/min, FiO_2_: 30%), and the CRP level increased to 28.1 mg/dL.

On Day 5, fever decreased to 37.0 °C and shortness of breath was aggravated. The patient developed ARDS (PaO_2_/FiO_2_: 108.0 mmHg). Chest radiography revealed increased haziness in both lower lung fields (Figure 1). The patient was intubated and a ventilator was applied.

On Day 6, the blood pressure decreased (60/48 mmHg), and septic shock was considered. After initiation of the inotropic agent, blood pressure returned to normal, and immunoglobulin administration via the venous route was initiated. The inotropic agent was discontinued on Day 7.

On Day 9, fever persisted at 38.3 °C, and cefepime and teicoplanin were added considering bacterial superinfection.

On Day 14, hypoxemia (PaO_2_: 54.1) was aggravated although high O_2_ was applied (FiO_2_: 90%). Chest radiography showed haziness in both lower lung fields, which was dominant in the left lower lung field. High-pressure mechanical ventilation was continued [positive end-expiratory pressure (PEEP):16 cmH_2_O, mode: pressure control mode]. Intravenous steroids were administered on the same day.

Ventilator weaning was attempted since Day 19 but was unsuccessful. On Day 28, the PaO_2_/FiO_2_ ratio was 134.4 mmHg and tracheostomy was performed. Laboratory examinations revealed normal levels of aspartate aminotransferase (AST) (27 U/L), alanine aminotransferase (ALT) (32 U/L), creatinine (0.9 mg/dL), elevated total bilirubin (1.42 mg/dL), CRP (3.46 mg/dL), and leukocytosis (16.72 × 10^9^/L). MSC transplantation was performed on that day and administered the next day.

Early adverse events, including allergic reactions, were not observed following MSC transplantation. Three days after MSC transplantation, a chest CT was performed, which showed ground-glass opacity (GGO) in both lungs, consolidation, and a cavity in the right upper lobe (Figure 2).

Five days after MSC transplantation, seizures with eyeball deviation were observed and anterior cerebral artery territory infarction was suspected on CT. Aspirin was then added to prevent further infarction. This was confirmed via magnetic resonance image.

Seven days after administration, the patient was mechanically ventilated at PEEP: 8 cmH_2_O and pressure-controlled mode (pressure setting:8 cmH_2_O) with a value of FiO_2_:0.5. PaO_2_/FiO_2_ was 129.1 mmHg, and O_2_ saturation was 94.4. Chest radiography showed slight regression of the GGO in both lower lung fields (Figure 3).

Fourteen days after MSC transplantation, the patient was mechanically ventilated at PEEP: 6 cmH_2_O and pressure support mode (pressure setting: 6 cmH_2_O) with values of FiO_2_:0.4. PaO_2_/FiO_2_ was 249.5 mmHg, and O_2_ saturation was 97.7. Chest radiography showed slight regression in the GGO in both lower lung fields.

The ventilator was weaned on Day 25 after the MSC transplantation. On Day 28 after the administration, high-flow nasal oxygen therapy was applied to the patient with values of FiO_2_: 0.4, and a flow rate of 30 L/min. PaO_2_/FiO_2_ was 279.4 mmHg, and O_2_ saturation was 98.2%.

Details of the method for respiratory support, chest radiography findings, arterial blood gas analysis results, and observed adverse events on the day of follow-up after MSC transplantation are described in Table 1.

Isolation was discontinued on Day 55 after transplantation because the RT-PCR results were negative. Chest CT performed on Day 106 showed decreased GGO in both lungs, decreased consolidation, and a cavity in the right upper lobe relative to the previous CT scan (Figure 4). The patient was transferred to the Department of Rehabilitation for respiratory rehabilitation and discharged 110 days after MSC transplantation. After discharge, the patient went to a nursing hospital and died due to an acute kidney injury of unknown origin 1 month after discharge from our hospital.

## 4. Discussion

No specific COVID-19 treatment is currently available. Although mild cases may require supportive therapy, severe cases have been managed using approved or experimental therapies. These include chloroquine, which is used for prophylaxis and treatment of malaria; lopinavir/ritonavir, which is a combination therapy for HIV; remdesivir and favipiravir; and tocilizumab (IL-6 antagonist). In accordance with the severity or stage of the infection, various combinations of these drugs have been introduced and used in the treatment guidelines of various countries [17,18,19].

Multiple aspects of the pathophysiology causing ARDS can be targeted by cell-based therapy, a potentially novel therapy. In recent years, cell-based therapies have been introduced into preclinical studies of ARDS. Several cell types, including embryonic stem cells, induced pluripotent stem cells, MSC, pulmonary epithelial progenitor cells, and endothelial progenitor cells, have been investigated as potential therapeutic candidates [20]. MSCs are of particular interest as possible candidates for ARDS therapy [21]. In particular, allogeneic MSCs were used in this study. Allogeneic MSCs have been used for clinical trials given their low immunogenicity due to the low expression rate of MHC class I [22]. Autologous MSCs may be better in terms of immunogenicity. However, it takes approximately 30 days from bone marrow collection to manufacturing autologous MSCs, whereas, in the case of allogeneic MSCs stored in the cell bank, the MSCs can be manufactured within 7 days. Therefore, allogeneic MSCs would be suitable for indications requiring rapid treatment including COVID-19-induced ARDS.

Recently, stem cell therapy has been used for critically ill patients with COVID-19. Several clinical trials using MSC, such as phase 1 clinical trials (NCT 04252118), phase 1/2 clinical trials (NCT 04288102), and phase 2 clinical trials (NCT 04269525), are being conducted. Leng et al. [12] reported no early or late adverse events and improved functional outcomes after MSC infusion in seven patients with COVID-19 pneumonia. Liang et al. [13] described critically ill patients whose vital, clinical, and laboratory findings returned to normal after MSC therapy.

Several mechanisms have been suggested to explain the efficacy of MSCs. MSCs stimulate tissue recovery and regeneration by secreting several paracrine substances with anti-inflammatory, immunomodulatory, angiogenesis-promoting, antifibrotic, antibacterial, and structural repair capabilities [14,23]. Notably, MSCs exhibit immunomodulatory properties by reducing T-cell growth, modulating B cell activities, and suppressing dendritic cell maturation [24]. Moreover, MSC reprogrammed macrophages by secreting prostaglandin E2 to boost interleukin-10 production in a model of septic lung inflammation [25]. In addition, Islam et al. [26] discovered that the protective effect of MSC against lung injury is mediated by mitochondrial translocation to the pulmonary alveoli.

MSCs were administered via the intravenous route in this study. Wu et al. demonstrated that the colonization and differentiation of BMSCs in the organ are mainly induced by the microenvironment in the injured organ [27]. MSCs have an inherent chemotaxis ability to hone in on sites of inflammation [28]. MSCs express the SDF-1 chemokine receptor [chemokine (C-X-C motif) receptor 4, CXCR4], while the SDF-1/CXCR4 biological axis stimulates the recruitment of progenitor cells to the site of tissue injury [29]. Therefore, MSCs administered via the intravenous route may affect the injured lung effectively due to this homing mechanism.

Here, we report the case of a patient with ARDS due to COVID-19 who was successfully treated with MSC transplantation. Although COVID-19 typically reaches its peak within 7–14 days and then improves, our patient experienced rapid disease progression and no recovery despite all supportive and pharmacological interventions, including chloroquine and lopinavir/ritonavir. After the administration of MSC, neither immediate nor delayed adverse events were detected. In addition, clinical symptoms, signs, and laboratory findings, including PaO_2_/FiO_2_ and O_2_ saturation, improved. A recent systematic review reported that cell-based therapy reduces mortality rates and improved recovery [30]. In particular, 12 of the 14 enrolled studies reported an increase in the PaO_2_/FiO_2_ ratio after stem cell infusion, 7 of the 14 enrolled studies reported an increase in oxygen saturation, and 15 of the 17 studies showed an improvement in lung images such as chest CT or chest radiography. Additionally, 11 of the 14 studies demonstrated discontinuation of oxygen support (intubation or ECMO) after cell infusion. Twenty-seven of the enrolled studies in the systematic review did not report any adverse events [30].

This case report has several limitations. As this is a report of a single case, it is impossible to draw conclusions about the treatment outcomes. Second, although 28 days post-treatment follow-up revealed no adverse effects, the study’s timetable did not allow us to investigate potential long-term side effects. Third, since several treatments were attempted after MSC transplantation, we cannot conclude that MSC transplantation has an absolute effect on improving patient outcomes.

Despite these limitations, this study is meaningful because the patient, showing no signs of recovery despite several treatment strategies, achieved favorable clinical outcomes after MSC transplantation. To our knowledge, this is the first report in South Korea of the effectiveness of MSC for patients with ARDS due to COVID-19.

## 5. Conclusions

The results of this case report suggest that intravenous injection of MSC derived from the bone marrow is safe and acceptable and can lead to favorable outcomes for critically ill COVID-19 patients. However, we cannot conclude that MSC transplantation for COVID-19-induced ARDS is effective and completely safe. Large randomized controlled trials are required to prove the therapeutic potential of MSC for the treatment of this disease.

## Figures and Tables

**Figure 1 medicina-58-01698-f001:**
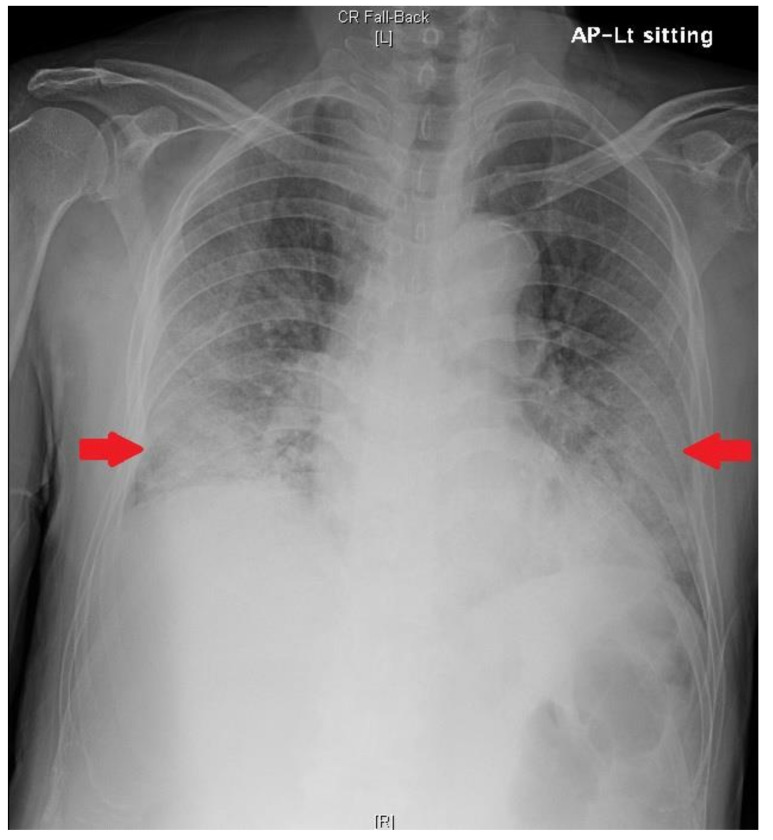
Chest radiograph obtained on day 6 of hospitalization shows haziness in both lower lung fields (red arrows).

**Figure 2 medicina-58-01698-f002:**
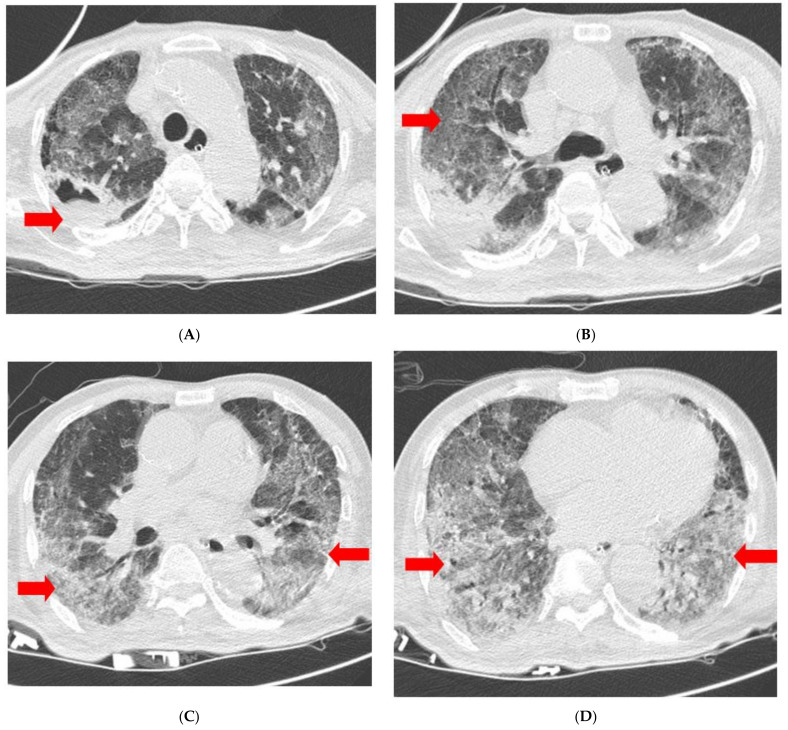
Chest CT scan obtained on Day 3 of MSC transplantation shows cavity (**A**) and pulmonary consolidation (**B**) at right upper lobe, and GGO at both lungs (**C**,**D**). Each finding is indicated by red arrowheads. CT, computed tomography; GGO, ground-glass opacity; MSC, Mesenchymal stem cell.

**Figure 3 medicina-58-01698-f003:**
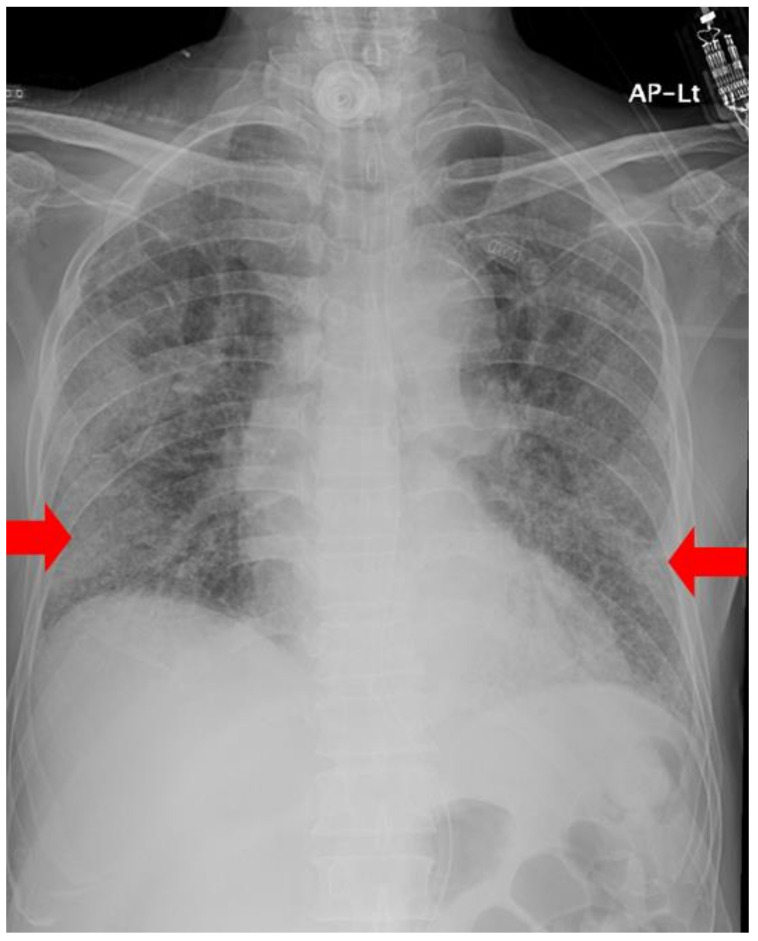
A chest radiograph obtained on Day 7 of MSC transplantation shows decreased haziness on both lower lung fields (red arrowheads).

**Figure 4 medicina-58-01698-f004:**
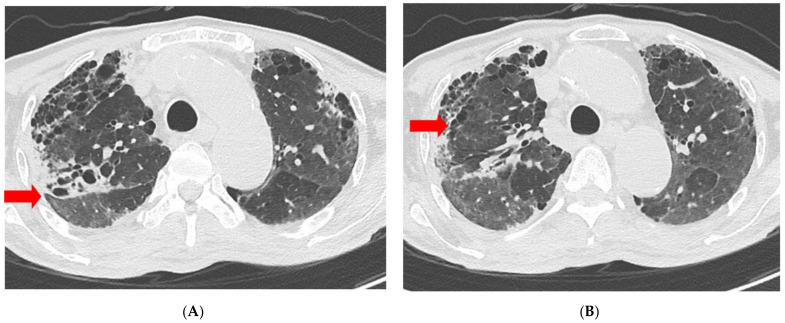
Chest CT scan obtained on Day 106 of MSC transplantation reveals absence of cavity (**A**) and pulmonary consolidation (**B**) at right upper lobe presented in the previous CT scan, and decreased GGO at both lungs (**C**,**D**) compared with previous CT scan. Each finding is indicated with red arrowheads. CT, computed tomography; GGO, ground-glass opacity; MSC, Mesenchymal stem cell.

**Table 1 medicina-58-01698-t001:** Respiratory support method, chest radiography findings, arterial blood gas analysis results, and adverse events on the day of follow-up after MSC transplantation.

Day after MSC Transplantation.	0	1	3	5	7	14	21	28
Respiratory support								
Mode	PCV	PCV	PCV	PCV	PCV	PSV	CPAP	HNFO
Pressure (cmH_2_O)	8	8	8	8	8	6 (pressure support)	-	-
Set frequency (breaths/min)	12	12	14	14	14	-	-	-
FiO_2_	0.5	0.5	0.7	0.5	0.5	0.4	0.4	0.4
PEEP (cmH_2_O)	8	8	12	12	8	6	5	-
Flow (L/min)	-	-	-	-	-	-	-	30
Chest radiography	GGO in BLLF	Increased GGO	Increased GGO	Decreased GGO	Decreased GGO	Decreased GGO	No interval change	No interval change
PaO_2_/FiO_2_ (mmHg)	137.4	135.4	145.8	159.0	129.1	249.5	314.2	279.4
O_2_ saturation (%)	93.2	92.7	87.3	96	94.4	97.7	99.0	98.2
Adverse event	No	No	No	No	No	No	No	No

MSC, mesenchymal stem cell; PCV, pressure-controlled ventilation; PSV, pressure support ventilation; CPAP, continuous positive airway pressure; HNFO, high-flow nasal oxygen therapy; PEEP, positive end-expiratory pressure; GGO, ground-glass opacity; BLLF, both lower lung fields.

## Data Availability

Not available.

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
