# Peer review of "Effectiveness of Mesenchymal Stem Cell Therapy for COVID-19-Induced ARDS Patients: A Case Report"

_medicina, 2022, doi:10.3390/medicina58121698_

Round 1

Reviewer 1 Report

In this clinical report, Lee et al. performed MSC transplantation as an attempt for COVID-19 patients. Considering that it happened at the beginning of COVID-19 pandemic, there was not specific treatment to tackle the ARDS. I do think this study was risky and gambling for the patient as an inflammatory environment may make the MSC to further intensify the complexity of the inflammation environment (e.g., Theranostics. 2022; 12(1): 207–231.) That’s why several studies attempted to use MSC-derived exosomes/extracellular vesicles instead of cell-based therapy. In this report, several parameters of the patient have been observed and measured continuously before and after the treatment, while no biological studies/biopsies were presented. The author did admit that it would be hard to draw any conclusions from this single patient treatment. I think it is still meaningful to publish these results as a good reference after addressing the following issues:

1. The author should add arrowheads to all CT and radiographs to indicate the significant changes in the lobes in the lungs. 

2. The author should explain and discuss how MSC may work to rescue the patient from the ARDS, such as how MSC may prefer to migrate to the inflammation sites and how do they work. Did the author extract blood to analyze the upregulation/downregulation of several markers that may involve in in this disease during the treatment?

3. What are the criteria for choosing a suitable batch of MSC? E.g., surface antigen matching with that of the patient, etc. Were autologous MSCs more suitable?

4. The ethical approval states that MSCs were i.v. injected once a day with 9 x10^7 number of cells. What is the rationale for choosing this scale for the treatment?

5. It would be nice if there is any follow-up work to track the healthy status of the patients nowadays. The patient was discharged 3-4 months post treatment (Mar 2020). Did he come back to the hospital to receive a regular body check these two years? Can the author add this information to the results?

Author Response

Reviewer 1

In this clinical report, Lee et al. performed MSC transplantation as an attempt for COVID-19 patients. Considering that it happened at the beginning of COVID-19 pandemic, there was not specific treatment to tackle the ARDS. I do think this study was risky and gambling for the patient as an inflammatory environment may make the MSC to further intensify the complexity of the inflammation environment (e.g., Theranostics. 2022; 12(1): 207–231.) That’s why several studies attempted to use MSC-derived exosomes/extracellular vesicles instead of cell-based therapy. In this report, several parameters of the patient have been observed and measured continuously before and after the treatment, while no biological studies/biopsies were presented. The author did admit that it would be hard to draw any conclusions from this single patient treatment. I think it is still meaningful to publish these results as a good reference after addressing the following issues:

Thank you for your comment.

1. The author should add arrowheads to all CT and radiographs to indicate the significant changes in the lobes in the lungs. 

Thank you for your comment. Red arrowheads have been added to indicate the findings for each figure.

2. The author should explain and discuss how MSC may work to rescue the patient from the ARDS, such as how MSC may prefer to migrate to the inflammation sites and how do they work. Did the author extract blood to analyze the upregulation/downregulation of several markers that may involve in in this disease during the treatment?

Thank you for your insightful comment. We added how MSCs migrate to the inflammation sites in the Discussion section as follows

MSCs were administered via intravenous route in this study. Wu et al. demonstrated that the colonization and differentiation of BMSCs in the organ are mainly induced by the microenvironment in the injured organ [27]. MSCs have an inherent chemotaxis ability to hone in on sites of inflammation [28]. MSCs express the SDF-1 chemokine receptor [chemokine (C-X-C motif) receptor 4, CXCR4], while the SDF-1/CXCR4 biological axis stimulates the recruitment of progenitor cells to the site of tissue injury [29]. Therefore, MSCs administered via intravenous route may affect the injured lung effectively due to this homing mechanism.”

And some description of the mechanisms of how MSC work has been added to the Discussion section.

Planned outcomes and follow-up day have been demonstrated in the Materials and methods section and Table 1. Other biologic markers outside our plan could not be shown in this case report because they were measured only as needed.

3. What are the criteria for choosing a suitable batch of MSC? E.g., surface antigen matching with that of the patient, etc. Were autologous MSCs more suitable?

Thank you for your insightful comment. Allogenic MSCs were used for clinical trials because MSCs have low immunogenicity due to the low expression rate of MHC class I. Autologous MSCs may be better in terms of immunogenicity. However, the reason for using allogeneic MSC in this study was that when using autologous MSC, it takes approximately 30 days from bone marrow collection to manufacturing MSCs, whereas in the case of allogeneic MSC stored in the cell bank, MSCs can be manufactured within 7 days. Hence, allogeneic MSC is suitable for indications requiring rapid treatment, such as COVID-19-induced ARDS. MSCs were collected from the bone marrow of donors who were negative for hepatitis B virus Ag, Hepatitis C virus Ab, HIV Ag & AB, HTLV-Ab, CMV IgG, IgM, and Treponema Ab. MSCs with a low risk of infection were identified through human virus test of MCB and appropriate MSCs were finally selected in consideration of cell phenotype, differentiation ability, cytokine secretion, and proliferation rates.

We added this content briefly in the discussion section as follows

In particular, allogeneic MSCs were used in this study. Allogeneic MSCs have been used for clinical trials given their low immunogenicity due to the low expression rate of MHC class I [22]. Autologous MSCs may be better in terms of immunogenicity. However, it takes approximately 30 days from bone marrow collection to manufacturing autologous MSCs, whereas in the case of allogeneic MSCs stored in the cell bank, the MSCs can be manufactured within 7 days. Therefore, allogeneic MSCs would be suitable for indications requiring rapid treatment including COVID-19-induced ARDS.”

4. The ethical approval states that MSCs were i.v. injected once a day with 9 x10^7 number of cells. What is the rationale for choosing this scale for the treatment?

Thank you for your comment. Pharmicell has produced Cellgram, which has been approved by the Korean Ministry of Food and Drug Safety. Cellgram AMI has been used for acute MI patients under approval by the Korean Ministry of Food and Drug Safety. Research on other products, such as Cellgram-AKI, CKD etc., is ongoing. The maximal dose of Cellgram is 9 x10^7 number of cells. Therefore, we requested approval from the Ministry of Food and Drug Safety by referring to the Cellgram dose for this study, and this was accepted.

5. It would be nice if there is any follow-up work to track the healthy status of the patients nowadays. The patient was discharged 3-4 months post treatment (Mar 2020). Did he come back to the hospital to receive a regular body check these two years? Can the author add this information to the results?

Thank you for your kind request. Actually, the initial plan of this study was to track progress until 30 days after MSC transplantation. The patient was transferred to a nursing hospital after discharge. Unfortunately, he passed away due to acute kidney injury (nontraumatic) 1 month after discharge. 

We have added a brief description of this point in the case presentation section to provide more complete case information as follows

“After discharge, the patient went to a nursing hospital and died due to an acute kidney injury of unknown origin 1 month after discharge from our hospital.”

Reviewer 2 Report

1) In the introduction, there is not sufficient information to support the use of MSCT in patients with ARDS or sever COVID-19. 

2) How many patients did the authors have recruited since the Ethical Committee has approved the protocol?

3) Why the authors only report one case and no more? This was the unique?

Author Response

1) In the introduction, there is not sufficient information to support the use of MSCT in patients with ARDS or sever COVID-19. 

 Thank you for your comment. We have added some content to address this concern as follows.

“Recently, some studies suggest that cell-based therapy provides therapeutic effects for critically ill patients with pneumonia due to COVID-19 [12,13]. The role of these stem cells may be supporting hematopoietic function, cell regeneration using multipotent cell differentiation ability, and immune modulation [14]. Therefore, it may be possible to suppress the overexpressed inflammatory response in ARDS patients.”

2) How many patients did the authors have recruited since the Ethical Committee has approved the protocol?

 Thank you for your query. A total of 10 patients from 3 institutions were enrolled in this study.

3) Why the authors only report one case and no more? This was the unique?

 Thank you for your comment. As stated in our response to comment 2, a total of

10 patients from 3 institutions were enrolled in this study. Ultimately, we decided to report on 1 patient treated at our institution, as approved by our institutional IRB due to difficulties in using data from other institutions.

Round 2

Reviewer 1 Report

The authors have addressed my comments appropriately.